# Short-Term Visual and Refractive Outcomes of Single-Step Transepithelial Photorefractive Keratectomy with Amaris 750S and SmartSurf^ACE^ in Myopia and Astigmatism: A 6-Month Follow-Up Study

**DOI:** 10.3390/life14101288

**Published:** 2024-10-11

**Authors:** Daiana-Andreea Margarit, Horia Tudor Stanca, Valeria Mocanu, Mihnea Munteanu, Suta Marius, Suta Gheorghita

**Affiliations:** 1Ophthalmology Department, “Victor Babes” University of Medicine and Pharmacy, 300041 Timisoara, Romania; 2Ophthalmology Department, “Carol Davila” University of Medicine and Pharmacy, 050474 Bucharest, Romania; 3Oftalmo Sensory-Tumor Research Center-ORL (EYEENT), 300041 Timisoara, Romania; 4Professor Munteanu Ophthalmology Center, 300092 Timisoara, Romania

**Keywords:** transepithelial PRK, SmartSurf^ACE^, myopia, astigmatism, refractive surgery, visual outcomes

## Abstract

Background: Single-step transepithelial photorefractive keratectomy (TPRK) is a modern refractive surgery technique that offers a no-touch approach for correcting myopia and astigmatism. This study aims to evaluate the short-term visual and refractive outcomes of TPRK in patients with myopia and astigmatism over a 6-month follow-up period. Methods: This retrospective cohort study included patients who underwent single-step TPRK using the Amaris 750S laser platform with SmartSurf^ACE^ and SmartPulse^®^ technologies, targeting a plano refraction. Procedures were performed with aspheric, non-wavefront-guided profiles, and outcomes were assessed postoperatively. Results: 96% of eyes achieved 20/20 or better uncorrected distance visual acuity (UDVA), with 98% reaching 20/25 or better, and 100% achieving 20/32 or better. UDVA was the same or better than preoperative corrected distance visual acuity (CDVA) in 96% of eyes, and no eyes lost two or more Snellen lines. Refractive outcomes showed strong precision, with 93% of eyes within ±1.00 D of the target. Astigmatism correction was accurate, with 100% of eyes having ≤1.00 D of astigmatism, and 80% achieving a precise astigmatism angle of error between −5° and 5°. Conclusion: Single-step TPRK with SmartSurf^ACE^ and SmartPulse^®^ technologies provides highly effective and predictable visual and refractive outcomes for myopia and astigmatism. The procedure consistently delivers precise corrections with minimal complications, making it a reliable option for refractive surgery.

## 1. Introduction

Refractive surgery has undergone significant evolution since the arrival of excimer laser technology [1,2], which enabled the correction of common vision problems such as myopia, hyperopia, and astigmatism [3,4]. Among the various refractive procedures, photorefractive keratectomy (PRK) emerged as a pioneering technique [5]. PRK involves the use of an excimer laser to reshape the cornea by ablating the anterior corneal stroma after removing the corneal epithelium [3,6,7]. Despite being one of the earliest methods introduced for refractive error correction, PRK remains relevant due to its safety profile, particularly in patients with thin corneas where procedures like laser-assisted in situ keratomileusis (LASIK) might be less suitable due to the need for creating a corneal flap, which could compromise corneal biomechanics and increase the risk of iatrogenic ectasia [8,9,10,11]. Over the years, LASIK gained popularity over PRK due to faster visual recovery and reduced postoperative discomfort. However, LASIK’s requirement for flap creation raised concerns regarding biomechanical stability, especially in patients with high myopia, leading to complications such as ectasia [12]. As a result, surface ablation techniques like PRK have regained attention, particularly with advancements in adjuvant therapies like mitomycin C (MMC) to reduce postoperative haze [13,14,15,16], a common complication of PRK in high myopia [17]. These developments led to the refinement of PRK techniques, making it a more viable option for a broader range of refractive errors [18].

Transepithelial photorefractive keratectomy (TPRK) is a more recent evolution of PRK that aims to address some of the limitations associated with traditional methods [19]. TPRK, often termed a “no-touch” technique, involves the simultaneous removal of the corneal epithelium and stromal ablation in a single step using an excimer laser [20,21,22]. This approach eliminates the need for mechanical or chemical removal of the epithelium, which has been associated with increased postoperative discomfort, risk of infection, and potential toxicity to limbal stem cells when alcohol-based debridement is used [12,18]. The development of TPRK has been facilitated by advancements in excimer laser technology, such as the Schwind Amaris platform [4]. This system utilizes advanced algorithms that consider variations in epithelial thickness across the corneal surface, allowing for more precise ablation profiles. In traditional PRK, the epithelial removal process was uniform, assuming a consistent thickness across the cornea. However, it is now well established that epithelial thickness is not uniform, leading to the risk of uneven ablation and suboptimal visual outcomes in conventional PRK [18]. TPRK’s ability to customize the ablation profile based on these variations represents a significant improvement, resulting in better visual outcomes, quicker recovery times, and reduced patient discomfort [17]. One of the specific advancements within TPRK is the SmartSurf^ACE^ technology, which further refines the procedure by integrating aberration-neutral profiles and an optimized epithelial removal process. SmartSurf^ACE^ also minimizes the thermal load on the cornea during ablation, further reducing the risk of postoperative complications such as haze and prolonged healing [11,19,23]. Despite these advancements, TPRK’s application in various refractive errors, particularly in myopia and astigmatism, requires further exploration to fully understand its long-term outcomes compared to other refractive surgeries like LASIK or small-incision lenticule extraction (SMILE) [23]. Moreover, given that TPRK eliminates the need for contact with the cornea, it could potentially offer a safer alternative for patients with thinner corneas or other corneal irregularities that make them unsuitable candidates for LASIK [23].

The aim of this study is to evaluate the short-term visual and refractive outcomes of single-step TPRK using the Schwind Amaris 750S with SmartSurf^ACE^ technology in patients with myopia and astigmatism over a six-month follow-up period. This research will contribute valuable insights into the role of TPRK in contemporary refractive surgery.

## 2. Materials and Methods

### 2.1. Design of the Study

This research was structured as a retrospective case series, carried out at the Ophthalmology Department of “Victor Babes” University of Medicine and Pharmacy in Timisoara, Romania. The study period ranged from January 2019 to June 2023. In keeping with the ethical principles outlined in the Declaration of Helsinki, the research focused on analyzing existing clinical records. Since this study did not involve new data collection, patient contact, or interventions, ethical approval was waived. This decision was based on the fact that the study used anonymized data, setting no risk to patient safety.

In terms of ethical practices, participants had provided digital informed consent during their initial clinical visits. This consent process clearly explained the study’s goals, procedures, and how their data would be used for academic purposes. By agreeing, participants gave permission for their data to be used in this retrospective study, ensuring transparency and ethical compliance throughout the research.

### 2.2. Patient Selection and Criteria

The collected data were comprehensive, including demographic information, visual acuity readings, refraction measurements, keratometry data, surgical details, and any recorded complications.

To be included in the study, patients had to meet the following criteria: (1) be at least 18 years old; (2) have a stable refractive error, defined as no change greater than 0.50 diopters within the last year; (3) have myopia no greater than −5.50 diopters and astigmatism no greater than −4.00 diopters; (4) have a healthy corneal topography with no signs of ectatic conditions; and (5) have a central corneal thickness of at least 500 µm to ensure a postoperative residual stromal bed of at least 300 µm.

Patients were excluded if they met any of the following conditions: (1) systemic conditions impairing wound healing, such as poorly controlled diabetes or autoimmune diseases; (2) pre-existing ocular conditions, including uncontrolled glaucoma or herpes simplex eye disease; (3) significant dry eye syndrome that could not be controlled medically; or (4) pregnancy or nursing due to potential changes in refraction.

### 2.3. Preoperative Assessments

Before undergoing the TPRK procedure, all patients went through a detailed preoperative evaluation to ensure both safety and the best possible surgical results.

Optometric Evaluation: (1) Subjective refraction was conducted to determine the patient’s distance refraction. This was further refined through cycloplegic refraction using an autorefractometer (KR8800, Topcon, Tokyo, Japan) and finalized with the Jackson cross-cylinder method; (2) ocular alignment was checked using a cover test with a translucent occluder (Optometric Promotion, Burgos, Spain); and (3) the near point of convergence was measured to assess the patient’s ability to maintain binocular single vision at close distances. Ophthalmologic Evaluation: (1) Corneal topography and keratometry were performed using the Sirius Topographer (CSO, Florence, Italy); (2) intraocular pressure and corneal biomechanics were assessed with a Topcon Tonometer (Topcon Medical Systems, Tokyo, Japan); (3) optical coherence tomography (OCT) was used to evaluate retinal health with Clarus OCT (Carl Zeiss Meditec, Jena, Germany); (4) anterior segment examination was performed using a slit lamp biomicroscope (LH-2000, Indo, Barcelona, Spain); and (5) a thorough fundoscopy was conducted after cycloplegic dilation using a binocular indirect ophthalmoscope (BIO) (Heine, Herrsching, Germany) to examine the internal structures of the eye.

### 2.4. Surgical Procedure

The TPRK surgeries were conducted using the Amaris 750S laser platform (SCHWIND eye-tech-solutions GmbH, Kleinostheim, Germany), operating at a frequency of 750 kHz. The optic zone was set at 6.75 ± 0.26 mm, with variations between 6.30 and 7.30 mm. High-energy fluence was set at 4.71 ± 0.30 mJ/cm^2^, and low-energy fluence was adjusted to 4.13 ± 0.55 mJ/cm^2^. Wavelengths were set at 655.62 ± 22.85 nm for high and 559.82 ± 21.90 nm for low. The procedure used aspheric, non-wavefront-guided profiles, and only proceeded when the remaining stromal thickness was confirmed to be over 300 microns.

Before surgery, patients followed a standardized regimen. They were instructed to use Moxifloxacin 0.5% eye drops (Vigamox, Alcon Laboratories, Fort Worth, TX, USA) three times daily, starting the day before surgery, to minimize infection risk. Additionally, Novesine 0.4% topical anesthetic drops (OmniVision, Puchheim, Germany) were applied three times for comfort during the surgery. The periocular area was thoroughly cleansed with a 5% povidone–iodine solution (Betadine, Avrio Health L.P., Stamford, CT, USA) to ensure sterility. The corneal surface was evenly cleaned with a balanced salt solution (BSS) to avoid any irregularities during laser application. The laser software calculated the optimal ablation profile based on each patient’s preoperative refractive error and corneal topography. Mitomycin C (0.02%, Kyowa-hakko Co. Ltd., Tokyo, Japan) was applied for 30 s to reduce the risk of postoperative haze by inhibiting fibroblast growth. The treated area was then thoroughly irrigated with BSS and dried.

### 2.5. Outcomes Measures

The outcome measures of this study included the following: (1) uncorrected distance visual acuity (UDVA) at 1, 3, and 6 months postoperatively; (2) corrected distance visual acuity (CDVA) at 1, 3, and 6 months, and comparison to preoperative CDVA; (3) spherical equivalent (SE) for refractive accuracy, with assessments of the correlation between attempted and achieved refraction; (4) astigmatism correction, including the analysis of refractive astigmatism magnitude and angle of error; (5) incidence of postoperative complications, specifically corneal haze; and (6) the safety index, based on changes in CDVA, and the efficacy index, based on postoperative UDVA.

### 2.6. Postoperative Care and Follow-up

Immediately After Surgery: Once the procedure was completed, patients were given several medications to prevent infection, control inflammation, and support healing. These included Moxifloxacin 0.5% eye drops (Vigamox, Alcon Laboratories, Fort Worth, TX, USA) to prevent infection, Bromfenac 0.09% eye drops (Yellox, Bausch & Lomb, Laval, QC, Canada) and Cyclopentolate 1% (OmniVision, Puchheim, Germany) to manage inflammation, and Tropicamide 1% (Bausch & Lomb, Rochester, NY, USA) as part of the postoperative regimen. The ocular surface was kept lubricated with Thealoz Duo (Laboratoires Théa, Clermont-Ferrand, France), and a Night & Day bandage contact lens (Alcon, Fort Worth, TX, USA) was placed to aid in epithelial healing. Additionally, Tobrex ointment (Tobramycin 0.3%, Alcon Laboratories, Fort Worth, TX, USA) was applied to provide further infection protection.

At Home: Patients were instructed to follow a strict medication routine at home. They continued using Moxifloxacin 0.5% eye drops four times daily for seven days. Tropicamide 1% was continued at four times daily for three days. Starting on the third postoperative day, Flumethasone 0.1% eye drops (Flumetol S, Santen, Osaka, Japan) were introduced with a gradual taper: four times daily for two weeks, then three times daily for two weeks, twice daily for two weeks, and finally once daily for two weeks. Bromfenac 0.09% eye drops were maintained twice daily for two weeks. Thealoz Duo was recommended four times daily for up to six months to ensure corneal hydration and comfort.

### 2.7. Statistical Analysis

Data analysis was performed using SPSS software version 29.0 (IBM Corporation, Armonk, NY, USA), following the guidelines of the Standard Charts for Reporting Refractive Surgery [24,25,26,27,28,29]. The Shapiro–Wilk test was used to check the normality of the data distribution. For non-parametric dependent variables, a Wilcoxon’s *t*-test was utilized. Statistical significance was determined with a *p*-value of less than 0.05, with results presented within a 95% confidence interval.

## 3. Results

The demographic and surgical characteristics of the 92 eyes from 46 patients included in the study are summarized in Table 1. These characteristics include patient age, gender distribution, corneal thickness, and key surgical parameters.

Visual Outcomes at 1 Month Postoperative

At 1 month postoperative, visual outcomes for the 92 eyes targeted for plano refraction were included. A cumulative analysis of UDVA revealed that 72% of eyes achieved 20/20 or better, 88% reached 20/25 or better, and 97% achieved 20/32 or better (Figure 1A). When comparing postoperative UDVA to preoperative CDVA, 72% of eyes had UDVA that was the same or better than their preoperative CDVA, and 88% were within one Snellen line of their preoperative CDVA (Figure 1B). In terms of changes in CDVA, 97% of eyes experienced no change, while 1% of eyes lost one Snellen line of CDVA. Importantly, only 1% lost two or more lines of CDVA (Figure 1C). The UDVA improved significantly from the preoperative mean value of 0.61 ± 0.24 logMAR to 0.05 ± 0.07 logMAR at 1 month postoperative. The Wilcoxon signed-rank test showed a statistically significant difference with Z = −8.30 and *p* < 0.001.

Visual Outcomes at 3 Months Postoperative

At 3 months postoperative, the visual acuity outcomes showed further improvement. A cumulative analysis indicated that 89% of eyes achieved 20/20 or better UDVA, with 96% of eyes achieving 20/25 or better, and 99% reaching 20/32 or better (Figure 2A). A comparison between UDVA and preoperative CDVA revealed that 89% of eyes had UDVA that was the same or better than their preoperative CDVA, and 96% of eyes were within one Snellen line of their preoperative CDVA (Figure 2B). In terms of CDVA changes, 97% of eyes experienced no change, 2% lost one Snellen line, and 1% lost two or more Snellen lines (Figure 2C). The UDVA continued to improve from 1 month postoperative with a mean value of 0.05 ± 0.07 logMAR to 0.02 ± 0.05 logMAR at 3 months postoperative. The Wilcoxon signed-rank test revealed a significant difference with *Z* = −4.39 and *p* < 0.001.

Visual and Refractive Outcomes at 6 Months Postoperative

At 6 months postoperative, visual outcomes continued to be stable and satisfactory. A cumulative analysis demonstrated that 96% of eyes achieved 20/20 or better UDVA, with 98% reaching 20/25 or better, and 100% achieving 20/32 or better (Figure 3A). When comparing UDVA to preoperative CDVA, 96% of eyes had UDVA that was the same or better than their preoperative CDVA, and 98% of eyes were within one Snellen line of their preoperative CDVA (Figure 3B). In terms of changes in CDVA, 97% of eyes experienced no change, 2% lost one Snellen line, and 1% of eyes lost two or more lines of CDVA (Figure 3C). There was a further improvement in UDVA from 3 months postoperative with a mean value of 0.02 ± 0.05 logMAR to 0.01 ± 0.03 logMAR at 6 months postoperative. This change was statistically significant as shown by the Wilcoxon signed-rank test with *Z* = −4.42 and *p* < 0.001. The efficacy index was set at 1.02 and the safety index was set at 1.01.

Refractive outcomes at 6 months also showed significant precision. The spherical equivalent (SE) analysis revealed a strong correlation between attempted and achieved refraction (R^2^ = 0.8635), with a mean value of 0.16 ± 0.54 D, indicating minimal overcorrection (Figure 3D). In terms of refractive accuracy, 65% of eyes were within ±0.50 D of the intended target, and 93% were within ±1.00 D (Figure 3E). Stability of SE over time was +0.16 D between preoperative and 6-month postoperative measurements (Figure 3F).

Refractive astigmatism outcomes also improved substantially. Postoperatively, 84% of eyes had ≤0.50 D of astigmatism, and 100% had ≤1.00 D, a marked reduction compared to preoperative levels (Figure 3G). The correlation between target-induced astigmatism vector and surgically induced astigmatism vector was strong (R^2^ = 0.7643), indicating accurate correction of astigmatism with minimal overcorrection or undercorrection (Figure 3H). Finally, the refractive astigmatism angle of error analysis showed that 80% of eyes had an angle of error between −5° and 5°, demonstrating precision in astigmatism correction (Figure 3I).

Of the 92 eyes examined, 8 (8.70%) experienced mild corneal haze following surgery. These cases were effectively treated using Dexamethasone 0.1% eye drops, which were applied four times daily during the first week and then gradually reduced over a period of four to six weeks, depending on the patient’s recovery. This treatment strategy was designed to inhibit the growth of myofibroblasts and limit collagen deposition, which are key contributors to haze formation.

## 4. Discussion

This study demonstrated favorable visual and refractive outcomes following surgery. Most patients achieved excellent visual acuity, with a considerable proportion reaching 20/20 vision or better. Refractive results were consistent and aligned closely with the targeted outcomes, showing minimal deviation. The incidence of postoperative complications was low, with only mild corneal haze reported in a small number of cases, which was successfully treated with standard medication. Overall, the procedure proved to be effective and safe, providing stable visual improvements over the follow-up period.

The visual and refractive outcomes from our study can be effectively compared with those reported in previous research [4,11,12,17,18,19,23,30,31,32], as summarized in Table 2. At 6 months postoperative, 96% of eyes in our study achieved 20/20 or better UDVA. These results are consistent with Luger et al. [30], who reported 88% of eyes achieving 20/20 or better UDVA, and Zhang et al. [11], who found that 98% of eyes reached this level. However, our results slightly exceed those of Baz et al. [4], who observed 75.8% of eyes achieving 20/20 UDVA. The differences could be attributed to variations in patient demographics or follow-up duration across studies. When it comes to refractive accuracy, our study showed a mean postoperative SE of 0.16 ± 0.54 D at 6 months, indicating minimal overcorrection. This is closely aligned with the outcomes reported by Luger et al. [30], who found a mean SE of 0.06 D, and Aslanides et al. [12], who reported 0.09 D. However, Baz et al. [4] noted a mean SE of −0.06 D, indicating a slight undercorrection in their cohort. The consistency in refractive outcomes between our study and those reported by Luger et al. [30] and Aslanides et al. [12] suggests that the surgical technique and technology used are effective in achieving precise refractive correction. The slight undercorrection observed in Baz et al. [4] may reflect differences in preoperative refraction levels or variations in surgical approach.

In terms of safety, our study observed an exceptionally low incidence of significant CDVA loss, with no eyes losing two or more lines of CDVA. This aligns with other studies [11,12,17,19,30], with 0% of eyes losing two or more lines. This consistency across studies reinforces the safety of TPRK using the Schwind Amaris 750S in preserving or improving visual acuity postoperatively. The incidence of corneal haze in our study was 8.7%, with all cases being mild and effectively managed with corticosteroid therapy. This is lower than the 17.7% ± 14.62 haze rate reported in the summary of other studies, which ranged from 0% to 37%. For example, Aslanides et al. [31] reported a haze rate of 37%, which was significantly higher than our findings. The differences in haze incidence could be due to variations in postoperative care protocols, surgical techniques, or patient characteristics across studies. Our lower haze rate highlights the effectiveness of our postoperative management protocol in minimizing this complication.

### 4.1. Limitations

While this study offers valuable insights into the short-term visual and refractive outcomes of single-step TPRK using the Amaris 750S and SmartSurf^ACE^ technology in treating myopia and astigmatism, there are several limitations that should be noted. First, the six-month follow-up period, while adequate for assessing early outcomes, does not provide information on the long-term stability of results or potential late-onset complications. A longer follow-up would be needed to evaluate the durability of the outcomes and the possibility of regression over time.

Second, although the sample size in this study was sufficient for initial analysis, it may not be large enough to generalize the findings to a broader population. A larger study with more participants could offer more robust data and help identify fewer common complications or variations in outcomes among different patient groups. Lastly, this study focused on clinical outcomes without examining patient-reported outcomes, such as quality of life or satisfaction with the procedure. Including these aspects in future research would provide a more comprehensive understanding of the overall impact of the surgery.

### 4.2. Future Lines of Research

Future research should focus on several key areas to further enhance the outcomes of TPRK and other refractive surgery techniques. First, studies exploring the impact of these procedures on corneal biomechanics are essential. Understanding how the cornea’s structural integrity changes post-surgery could help in preventing complications like ectasia. Comparative studies are also needed to evaluate different epithelial removal techniques, such as mechanical, alcohol-assisted, and transepithelial methods. These comparisons can help identify the most effective and safest approaches for different patient profiles. Research into wavefront-guided treatments is another key area. By investigating how these advanced techniques affect higher-order aberrations, we can better tailor procedures for patients with complex refractive errors.

Additionally, exploring novel epithelial removal technologies in TPRK holds promise for improving visual outcomes and speeding up recovery times. Evaluating its effectiveness across a range of refractive errors will be crucial. For patients with high myopia, further studies are needed to determine whether TPRK can match or surpass traditional methods like LASIK in terms of safety and efficacy. Finally, the use of SmartSurf^ACE^ technology in correcting residual refractive errors post-SMILE is an emerging area that warrants further investigation. Validating its outcomes could expand its application in other refractive surgery techniques.

### 4.3. Practical Applications

The results of this study provide valuable insights that can be directly applied in the daily practice of refractive surgeons. The demonstrated safety and effectiveness of single-step TPRK for improving visual and refractive outcomes reinforce the reliability of this technique, particularly for patients with mild to moderate myopia and astigmatism. Surgeons can confidently consider TPRK as a viable alternative to traditional PRK methods, especially for cases where minimizing epithelial trauma is a priority. By incorporating TPRK into their practice, refractive surgeons can offer a streamlined and efficient procedure, potentially reducing the need for more invasive techniques. The consistency in achieving favorable visual acuity outcomes, as demonstrated in this study, supports the use of TPRK as a reliable option for patients seeking refractive surgery.

## 5. Conclusions

This study demonstrates that single-step TPRK is a safe, effective, and precise procedure for correcting myopia and astigmatism. The consistent visual and refractive outcomes at 1, 3, and 6 months postoperatively, with minimal complications, highlight the procedure’s reliability. These results support TPRK as a strong alternative to traditional refractive surgeries, offering predictable outcomes that can be confidently applied in clinical practice.

## Figures and Tables

**Figure 1 life-14-01288-f001:**
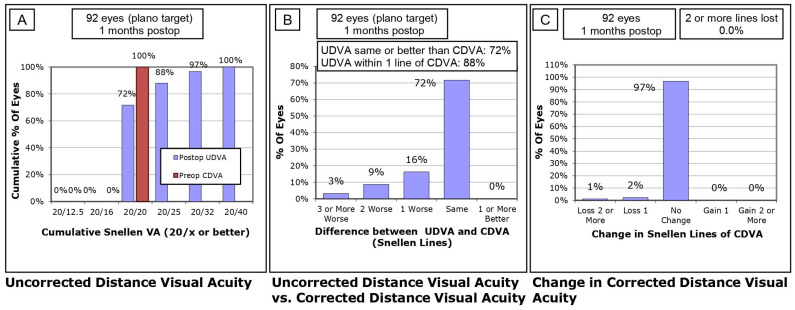
One-month postoperative visual acuity outcomes targeted for plano refraction. (**A**) Cumulative analysis of uncorrected distance visual acuity (UDVA) at 1 month postoperative. (**B**) Comparison of postoperative UDVA to preoperative corrected distance visual acuity (CDVA). (**C**) Changes in corrected distance visual acuity (CDVA) postoperatively.

**Figure 2 life-14-01288-f002:**
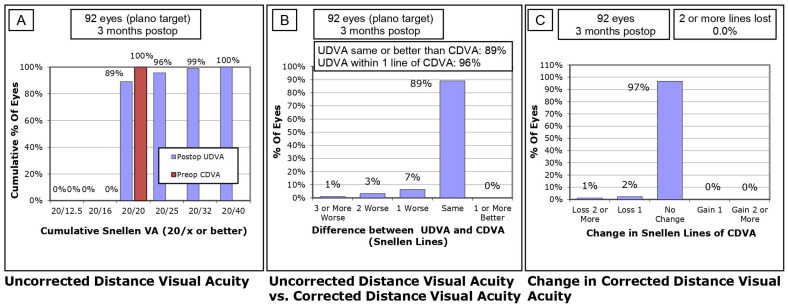
Visual acuity outcomes at 3 months postoperative for plano refraction. (**A**) Cumulative analysis of uncorrected distance visual acuity (UDVA) at 3 months postoperative. (**B**) Comparison of postoperative UDVA to preoperative corrected distance visual acuity (CDVA). (**C**) Changes in CDVA postoperatively.

**Figure 3 life-14-01288-f003:**
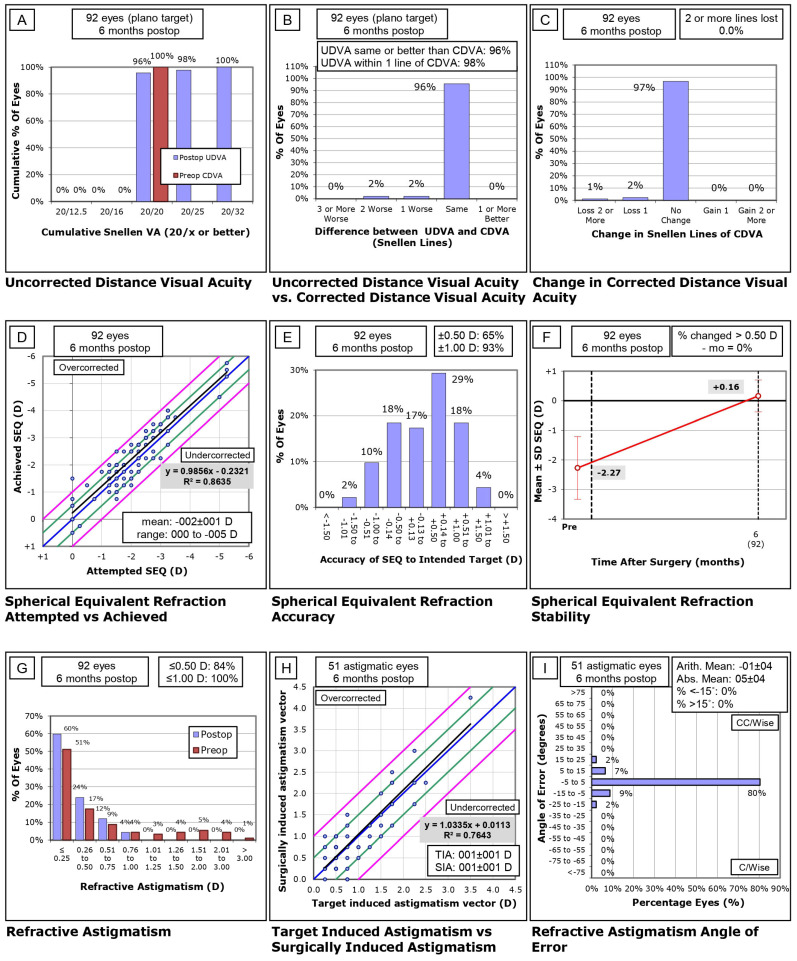
Visual and refractive outcomes at 6 Months postoperative for plano refraction. (**A**) Cumulative analysis of uncorrected distance visual acuity (UDVA) at 6 months postoperative. (**B**) Comparison of postoperative UDVA to preoperative corrected distance visual acuity (CDVA). (**C**) Changes in CDVA postoperatively. (**D**) Spherical equivalent (SE) analysis showing correlation between attempted and achieved refraction. (**E**) Refractive accuracy in terms of deviation from the intended target. (**F**) Stability of SE over time. (**G**) Refractive astigmatism outcomes postoperatively. (**H**) Correlation between target induced astigmatism and surgically induced astigmatism. (**I**) Refractive astigmatism angle of error analysis.

**Table 1 life-14-01288-t001:** Demographic and surgical parameters of the study population.

Parameter	Value
Mean Age (years)	29.02 ± 6.09 (Range: 20 to 53 years)
Gender Distribution (Male)	63%
Ethnicity (Caucasian)	100%
Central Corneal Thickness (µm)	549.52 ± 24.58 (Range: 506 to 622 µm)
Ablation Depth (µm)	98.63 ± 15.82 (Range: 71.40 to 159.0 µm)
Residual Stromal Bed Thickness (µm)	450.91 ± 30.81 (Range: 361.00 to 519.00 µm)
Surgical Duration (seconds)	38.89 ± 9.80 (Range: 25 to 73 s)

**Table 2 life-14-01288-t002:** Summary of Studies on visual and refractive outcomes following transepithelial PRK using Schwind Amaris 750S.

Author	Year	Platform	Eyes	Patients	Age	Follow-up (Months)	Preoperative Refraction (D)	Efficacy ^1^	Safety ^2^	Last Visit SE (D)	Haze (% Rate)
Luger et al. [30]	2012	750S	33	33	34.0 ± 9.0	12	−4.06 ± 2.11	88.0	0	0.06 ± 0.23	NR
Baz et al. [4]	2013	750S	33	19	31.4 ± 8.4	6	−1.76 ± 1.46	75.8	6	−0.06 ± 0.25	18.2
Aslanides et al. [31]	2014	750S	41	27	27 (6)	12	−7.89 (1.24)	77.1	0	−0.10 ± 0.34	37.0
Aslanides et al. [12]	2017	750S	89	45	29.81 (0.52)	6	4.33 (2.03)	87.5	0	0.09 ± 0.19	0
Bakhsh et al. [17]	2018	750S	100	50	28.3 ± 6.3	6	−3.15 ± 1.59	100.0	0	−0.05 ± 0.33	NR
Zarei-Ghavanati et al. [32]	2019	750S	40	40	31.2 ± 8.7	6	−3.98 ± 2.06	NR	NR	NR	32.5
Zhang et al. [11]	2020	750S	85	46	25.6 ± 6.1	12	−7.59 ± 0.84	98.0	0	−0.05 ± 0.39	NR
Alasmari et al. [18]	2021	750S	84	42	28.4 ± 6.4	6	−1.75 (0.50)	100.0	NR	0.00 (0.20)	6
Kaluzny et al. [19]	2021	750S	48	39	31 (34)	36	−0.62 ± 0.78	71.0	0	−0.17 ± 0.26	12.5
Li et al. [23]	2023	750S	408	212	26.6 ± 6.0	24	−6.80 ± 1.18	NR	NR	0.03 ± 0.42	NR

^1^ Efficacy was measured as percentage of eyes with UDVA ≥ 20/20, ^2^ safety was measured as percentage of eyes with loss of CDVA ≥ 2 lines. Diopters, UDVA: uncorrected distance visual acuity, CDVA: corrected distance visual acuity, SE: spherical equivalent, NR: not reported. Data were presented as mean ± deviation or median (interquartile range).

## Data Availability

The raw data supporting the conclusions of this article will be made available by the authors on request.

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
