# Peer review of "Short-Term Visual and Refractive Outcomes of Single-Step Transepithelial Photorefractive Keratectomy with Amaris 750S and SmartSurfACE in Myopia and Astigmatism: A 6-Month Follow-Up Study"

_life, 2024, doi:10.3390/life14101288_

Round 1

Reviewer 1 Report

Comments and Suggestions for Authors

Thank you for submitting the article "Short-Term Visual and Refractive Outcomes of Single-Step Transepithelial Photorefractive Keratectomy with Amaris 750S and SmartSurfACE in Myopia and Astigmatism: A 6-Month Follow-Up Study" to MDPI life

The article provides a retrospective analysis of refractive surgery outcomes from a single centre, utilising the Amaris 750S laser platform with SmartSurfACE and SmartPulse technologies. The study offers valuable insights into the clinic's laser refractive surgery practice, providing clinical data with a six-month follow-up period.

Please consider the following comments on the article for further revision: 

Scope: MDPI life is wide is scope, however, it is debatable and left to the authors to decide if it is the most adequate journal for the presentation of refractive surgery results. 

L194: please remove the template sentence. 

L197: “demonstrated favourable results” Please avoid qualitative statements in the results section. 

Comment on figures & legends: Please avoid legends in the figures (statistics output) and below the figure again. Instead refer to Figure 1A etc. 

Please confirm whether there is no relevant financial, relational or other conflict of interest. 

Comments on the Quality of English Language

English generally fine, minor corrections. 

Author Response

Reviewer 1

#RV1_0: Thank you for submitting the article "Short-Term Visual and Refractive Outcomes of Single-Step Transepithelial Photorefractive Keratectomy with Amaris 750S and SmartSurfACE in Myopia and Astigmatism: A 6-Month Follow-Up Study" to MDPI life. The article provides a retrospective analysis of refractive surgery outcomes from a single centre, utilizing the Amaris 750S laser platform with SmartSurfACE and SmartPulse technologies. The study offers valuable insights into the clinic's laser refractive surgery practice, providing clinical data with a six-month follow-up period.

#AU1_0: Thank you for your valuable feedback and for considering our manuscript for publication in Life. We are pleased that our retrospective analysis and clinical data provide meaningful insights into refractive surgery outcomes using the Amaris 750S platform with SmartSurfACE and SmartPulse technologies.

#RV1_1: Please consider the following comments on the article for further revision:

Scope: MDPI life is wide is scope, however, it is debatable and left to the authors to decide if it is the most adequate journal for the presentation of refractive surgery results.

#AU1_1: Thank you for your comment regarding the scope of Life. We would like to clarify that our submission is aligned with the journal’s Special Issue on “New Diagnostic and Therapeutic Developments in Eye and Systemic Diseases,” which explicitly includes refractive surgery as a relevant topic. Our study contributes valuable insights into modern refractive surgery techniques and their clinical outcomes, making it a suitable fit for the Special Issue's focus on innovative treatments in ophthalmology.

#RV1_2: L194: please remove the template sentence.

#AU1_2: We have removed the template sentence as requested and revised the manuscript accordingly. Thank you for bringing this to our attention.

#RV1_3: L197: “demonstrated favourable results” Please avoid qualitative statements in the results section.

#AU1_3: Thank you for your feedback. We have revised the results section to remove the qualitative statement "demonstrated favorable results"

#RV1_4: Comment on figures & legends: Please avoid legends in the figures (statistics output) and below the figure again. Instead refer to Figure 1A etc.

#AU1_4: Thank you for your feedback. We will remove the legends within the figures and ensure that all references to the figures are made in the main text using labels such as "Figure 1A" instead. This will streamline the presentation and avoid redundancy.

#RV1_5: Please confirm whether there is no relevant financial, relational or other conflict of interest.

#AU1_5: We confirm that there are no relevant financial, relational, or other conflicts of interest associated with this manuscript, or the research conducted. Additionally, we have included a conflict of interest declaration statement at the end of the manuscript to clarify this further.

Reviewer 2 Report

Comments and Suggestions for Authors

The authors showed impressive study that short term visual and refractive outcomes of single-step TPRK using the Amaris 750S and SmartSurfACE technology in treating myopia and astigmatism. Totally author’s data are wonderful due to six-month follow-up period is too short.

Only positive results let us consider their technology will not succeed during long-term period.

I think the author’s study should be shown as long-term follow-up period. 

Author Response

Reviewer 2

#RV2_0: The authors showed impressive study that short term visual and refractive outcomes of single-step TPRK using the Amaris 750S and SmartSurfACE technology in treating myopia and astigmatism. Totally author’s data are wonderful due to six-month follow-up period is too short.

#AU2_0: Thank you for your positive feedback. We agree that while the six-month follow-up provides valuable short-term data, it is indeed brief for assessing long-term success. As mentioned in the limitations, we plan extended studies to evaluate long-term outcomes, including stability, potential regression, and late-onset complications.

#RV2_1: Only positive results let us consider their technology will not succeed during long-term period. I think the author’s study should be shown as long-term follow-up period

#AU2_1: Thank you for your insightful comment. While our study reports positive short-term results, we recognize that long-term success requires ongoing evaluation. This is why we have clearly stated in the limitations section of the manuscript that the six-month follow-up period is a short-term assessment, and longer follow-up is necessary to determine the technology's long-term stability, the potential for refractive regression, and the emergence of any late-onset complications.

We fully agree that the long-term efficacy of the technology will require further study, and we plan to address this in future research by extending the follow-up period to assess the durability of the outcomes over time.

Reviewer 3 Report

Comments and Suggestions for Authors

Revise the study design. The methodology is not consistent with a cohort study. 

Clarify the exclusion criteria of this retrospective study 

Include outcome measures in the methods

Comments on the Quality of English Language

None

Author Response

Reviewer 3

#RV3_1: Revise the study design. The methodology is not consistent with a cohort study.

#AU3_1: Thank you for your comment regarding the study design. We recognize the need for consistency in describing the methodology. While the study is currently referred to as a "retrospective cohort study," we acknowledge that the retrospective nature of analyzing pre-existing clinical records might be more accurately described as a "retrospective case series" rather than a cohort study. We have revised the manuscript to reflect this and updated the design description accordingly.

#RV3_2: Clarify the exclusion criteria of this retrospective study

#AU3_2: Thank you for your comment. We have clarified the exclusion criteria to ensure transparency.

To be included in the study, patients had to meet the following criteria: (1) be at least 18 years old; (2) have a stable refractive error, defined as no change greater than 0.50 diopters within the last year; (3) myopia no greater than -5.50 diopters and astigmatism no greater than -4.00 diopters; (4) have a healthy corneal topography with no signs of ectatic conditions; (5) have a central corneal thickness of at least 500 µm to ensure a postoperative residual stromal bed of at least 300 µm.

Patients were excluded if they met any of the following conditions: (1) systemic conditions impairing wound healing, such as poorly controlled diabetes or autoimmune diseases; (2) pre-existing ocular conditions, including uncontrolled glaucoma or herpes simplex eye disease; (3) significant dry eye syndrome that could not be controlled medically; (4) pregnancy or nursing due to potential changes in refraction.

#RV3_3: Include outcome measures in the methods

#AU3_3: Thank you for your suggestion. We have now included a description of the outcome measures in the methods section.

“The outcome measures of this study included the following: (1) Uncorrected distance visual acuity (UDVA) at 1, 3, and 6 months postoperatively, (2) Corrected distance visual acuity (CDVA) at 1, 3, and 6 months, and comparison to preoperative CDVA,  (3) Spherical equivalent (SE) for refractive accuracy, with assessments of the correlation between attempted and achieved refraction, (4) Astigmatism correction, including the analysis of refractive astigmatism magnitude and angle of error, (5) Incidence of post-operative complications, specifically corneal haze and (6) Safety index, based on changes in CDVA, and efficacy index, based on postoperative UDVA.”